# Numerical Investigation of Erosion Wear Characteristics of Hydraulic Spillway

**Cong Zhang** [1,2], **Yuqi Zhang** [1,2], **Huadong Zhao** [1,2,*], **Mao Wang** [1,2] and **Tongtong Wang** [1,2]

1   College of Mechanical and Power Engineering, Zhengzhou University, Zhengzhou 450001, China; zc15872417933@163.com (C.Z.); 15839127287@163.com (Y.Z.); wangmaoyouxiang@163.com (M.W.); tongtongw19970228@163.com (T.W.)
2   Intelligent Manufacturing Research Institute of Henan Province, Zhengzhou 450001, China
*   Correspondence: huadong@zzu.edu.cn

**Abstract:** There are many sand-laden waters in China, and the wear of hydraulic structures caused by sand-laden water diversion has been paid more and more attention. Taking the spillway of a reservoir as the research object, the numerical model of erosion wear caused by sediment-laden particle flows on the spillway was established by using the computational fluid dynamics (CFD) method, VOF (Volume of Fluid) multiphase flow model and DPM (Discrete Phase Model). Through the simulation analysis of the spillway's overall erosion, the distribution of the spillway erosion wear was obtained. Then, according to the main wear parts, the single variable, such as sediment diameter, sediment concentration, flow velocity and gate opening degree, was changed to study the erosion amount of the spillway and the distribution law of the spillway erosion parts. The results show that the main erosion sites of the spillway are at the bottom of the gate chamber and the middle section of the spillway. The maximum erosion increases linearly with the increase in sediment concentration. With the increase in sediment concentration, the sediment concentration changed from $1 \text{ kg/m}^3$ to $6 \text{ kg/m}^3$, and the maximum erosion of the spillway increased from $2.58 \times 10^{-7} \text{ kg/m}^2$ to $1.53 \times 10^{-6} \text{ kg/m}^2$. The erosion at the bottom of the spillway and gate leaf increases first and then decreases with the increase in sediment diameter and reaches the maximum value when the particle size is 0.002 mm. The erosion at the bottom of the spillway and the gate leaf increases with different growth trends as the flow velocity increases, when the flow velocity increases from 2 m/s to 9 m/s and the maximum erosion amount at the bottom of the spillway increases from $3.66 \times 10^{-7} \text{ kg/m}^2$ to $1.14 \times 10^{-6} \text{ kg/m}^2$, and the maximum erosion of the gate leaf increased from $1.66 \times 10^{-8} \text{ kg/m}^2$ to $8.98 \times 10^{-6} \text{ kg/m}^2$. The erosion amount at the bottom of the spillway increases with the increase in the gate opening between 0 and 3 m and tends to be stable when the gate opening is greater than 3 m. The maximum erosion position moves to the rear part of the spillway with the change in the gate opening. The change in the gate opening has no obvious effect on the erosion amount of the gate leaf but only changes the area of the gate erosion part. Thus, the erosion wear distribution of spillway under different work conditions is summarized, and the qualitative study between the erosion wear and the distribution of sediment diameter, sediment concentration, flow velocity and gate opening degree is made.

**Keywords:** hydraulic structures; erosive wear; computational fluid dynamics (CFD); VOF (Volume of Fluid); multiphase; DPM (Discrete Phase Model)

## 1. Introduction

China is rich in water resources; with the full use of water resources in China, a variety of hydraulic constructions are under-constructed. China is a country with many sediment-laden rivers, and the Yellow River basin is famous for its sediment. The erosion and wear of hydraulic structures caused by sand flow is a common disease of hydraulic structures [1–3]. According to the survey, 70% of hydraulic structures in sandy rivers suffer

from wear and damage caused by high-speed sediment-laden flow. For example, since Yuzixi [4] Hydropower Station in Sichuan Province was put into operation in May 1986, the reinforced concrete and steel plate lining in the sand drainage tunnel cannot withstand the impact wear caused by flood discharge and sand discharge every year. Four years after the completion and operation of Ertan Hydropower Station [5], the local abrasion pit of the No.1 spillway tunnel was 20~30 mm deep. After working for 8666.34 h in total, serious abrasion pits appeared at the bottom of the curved gate of Xiaolangdi No.1 Sand Drain Tunnel, which seriously affected the safe operation of the gate. One year after the construction and operation of Daquke Hydropower Station, the erosion damage occurred in the sand scouring tunnel, the steel lining of the bottom plate of the working gate was seriously damaged, and the water sealing bolt on the bottom edge of the working gate was worn and fell off.

At present, the erosion wear of hydraulic structures is mainly studied by scouring experiments with various scour experimental instruments. Hu et al. [6] concluded through experimental studies that the erosion rate of concrete is basically linear with the increase in sediment concentration, and the change in velocity at a low angle scour has a great influence on the erosion rate. A.W. Monber [7] and Yin [8] improved the traditional erosion wear equipment, and conducted experiments with the improved equipment, and found that the erosion wear amount of concrete increased exponentially with the increase in jet velocity. Xiong [9] carried out the erosion wear experiment on Q345 steel and concluded that the erosion amount increased with the equivalent increase in particle size and concentration. With the change in impact angle, the wear quantity increases first and then decreases with the change in impact angle and reaches the maximum value at 30°. Leporini et al. [10] carried out a preliminary experimental study on the three-phase flow (air–water–sand) in a horizontal pipe and introduced the application of the sand–liquid model in the literature. The results show that sand transport characteristics and critical deposition rate are closely related to gas–liquid two-phase flow state and sand concentration.

With the development of computers, numerical simulation is widely used in erosion simulation [11,12], such as black water pipelines, bifurcation pipelines of hydropower stations and oil–gas pipelines [13,14]. Qiao [15] used the computational fluid dynamics method to numerically simulate the erosion wear behavior and mechanism of solid–liquid two-phase pipeline in coal gasification wastewater treatment system and optimized the bend of black water pipeline. Farokhipour et al. [16] simulated and analyzed the influence rules of fluid velocity, particle diameter and solid concentration on pipeline erosion and wear caused by multiphase flow mud. The simulation study by Wang et al. [17] shows that there are two serious erosion positions in the oil delivery elbow, and the layout of the elbow and the particle size will significantly change the serious erosion positions of the elbow. Huang etc. [18], on the basis of previous research, put forward a theoretical calculation model, which is the calculation of the solid particles in the mud flow of tube wall erosion situation, and found that the erosion rate and average mud speed is very closely related, and the pipe diameter and fluid viscosity hedge the impact of corrosion rate is lesser, and the results also show that there is an exponential power relationship between erosion rate and average mud velocity, pipe diameter, particle size and fluid viscosity. Zhang [19] studied the erosion wear mechanism of tee tubes in-depth based on CFD; his research showed that under the confluence effect, the higher the gas flow rate in the tee, the lower the pressure, and the more likely the section is to suffer severe erosion. Based on the DPM erosion prediction model, Zhong et al. [20] studied the erosion wear of a new high-pressure manifold quick connection device on the surface of some well sites. Hashemisohi et al. [21] studied the bubbling characteristics and separation of multi-component particle mixtures in fluidized beds by using a dense discrete phase model combined with particle flow mechanics theory and experiment. Chen et al. [22] took the vertical grinding machine as the research object, applied the method of combining experiment and simulation, solved the particle motion trajectory with the bidirectional coupling calculation method of discrete phase model DPM and analyzed the mutual coupling effect of two-phase flow in the

vertical grinding flow field and the particle classification and screening characteristics from the aspects of velocity, pressure and discrete phase distribution. In terms of numerical simulation and experimental research on turbine wear, Liao et al. [23] carried out full flow channel numerical simulation on a Francis turbine by using VOF model and DPM model, and analyzed the sediment distribution and solid–liquid two-phase velocity on runner blade surface under different working conditions, which provided a theoretical basis for the transformation of turbine in this power station. Hong [24] established a CFD-DPM model to investigate the erosion of 90° elbow in a shale gas field under gas–solid two-phase flows, His findings include the influence of six factors on the maximum amount of erosion which including gas velocity, mass flow rate of sand particles, particle diameter, shape coefficient of sand particles, pipeline diameter, elbow radius of curvature. Zhu and Qi [25] used a CFD-DPM method which is based on Euler–Lagrange method and erosion model to study the flow erosion of sand-bearing oil flow in the U-bend. The finding shows that the maximum erosion part occurs on the outer elbow, the lower surface of the elbow and the downstream pipe. Zhang et al. [26] carried out numerical simulation and sediment wear test on the turbine of Hatta Hydropower Station, and predicted the sediment wear of the turbine, which provided an important reference for the anti-wear design and operation of the turbine of Hatta Hydropower Station. Si et al. [27] applied the wear model and DPM model to simulate the stay vane of a certain type of turbine, and the wear rate on the surface of the guide vane of the numerical simulation results are compared with the experimental data, the forecast and verified the turbine guide vane fixed under the rated conditions of wear situation of fixed for the turbine guide vane abrasion in the late design provides reference basis.

As a large hydraulic conservancy structure, the operating environment of the spillway is relatively complex; there are many factors that affect the sand flow, such as sand diameter, concentration, flow rate, gate opening and so on. It is difficult, through the prototype observation and in-situ experiment methods, to study the erosion wear characteristics.

The numerical simulation method can be used to systematically and comprehensively study the influence of different factors on the erosion of spillways, and the actual erosion situation of spillways can also be used to verify the accuracy of the numerical simulation. According to the above discussion, this paper adopts the method of numerical simulation, combined with the VOF model and DPM, and takes the spillway of a reservoir in the middle reaches of the Yellow River as the research object to analyze the influence of different factors on the erosion wear of the spillway. This will lay a foundation for the intelligent operation and maintenance of hydraulic engineering and provide important guiding significance, and at the same time, it will provide basic ideas and general methods for the erosion research of hydraulic structures.

The main research points of this paper include: (1) Summing up the theoretical basis of spillway erosion and establishing the numerical model of spillway erosion; (2) The spillway fluid zone erosion was simulated to analyze the main locations of spillway erosion, and the accuracy of simulation results was verified by combining field conditions and relevant actual cases; (3) Change single variables such as sediment particle size, sediment concentration, inlet velocity and gate opening degree to analyze the influence of single variable on spillway erosion and its mechanism. There is a brief introduction to the engineering background of the spillway, and a summary of the numerical basis of the numerical model of spillway erosion is provided in Section 2, then the finite model of the spillway was established. In Section 3, the distribution of wear on the spillway is specific and, the influence of different factors on the spillway was further studied. Finally, the main content of this paper is summarized in Chapter 4.

## 2. Numerical Model

### 2.1. Engineering Background

Taking the spillway of a reservoir as the research object, the spillway is composed of the front section, the gate chamber section, the middle section and the rear section. The gate

chamber section is a three-hole curved gate structure, and these curved gates are separated by the gate pier. The bottom and wall of the spillway are formed by concrete pouring, and the gate material is Q435B. The size and structure of the spillway are shown in Figure 1.

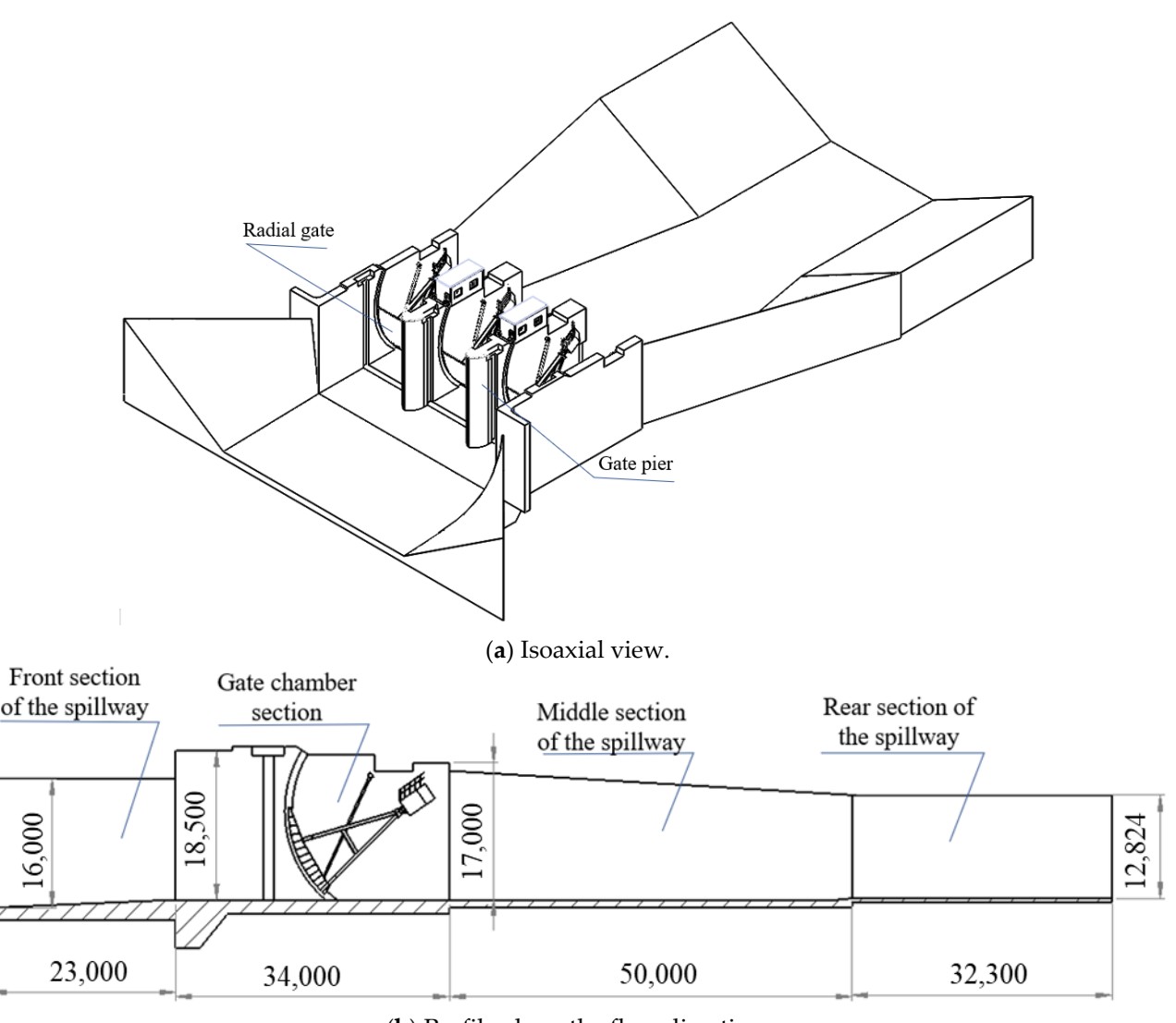

(**a**) Isoaxial view.

(**b**) Profile along the flow direction.

**Figure 1.** Spillway structure diagram.

The reservoir is located in the tributary of the Yellow River, with an average sediment concentration of 3.2 kg/m$^3$. More than 90% of the sediment is concentrated in the flood season, and the water level in the flood season is 6.5 m, so the flood season of the spillway is seriously eroded by sediment. According to the relevant literature [28–30], the particle size of sediment in the Yellow River Basin ranges from 0.008 to 0.034 mm, and the average density is $2.73 \times 10^3$ kg/m$^3$. In the simulation calculation, the influence of particle shape on spillway erosion is not considered, and the sand particles are assumed to be homogeneous spheres.

### 2.2. Numerical Model

#### 2.2.1. VOF Model Theory

When the spillway is discharged, the continuous phase is water and air. The VOF model is a surface tracking method based on a fixed Euler grid. The VOF model can track the phase interface of each computing unit by introducing volume fraction. The VOF

model is used to track the interface between water and air, and its continuity equation is expressed as [31]:

$$\frac{\partial \alpha_q}{\partial t} + v \cdot \nabla \partial_q = \frac{S_{\partial q}}{\rho_q} \tag{1}$$

where, $q$ represents the corresponding $q$ phase, $\alpha$ represents the volume fraction, $v$ represents the velocity in the system region, $S_{aq}$ represents the mass source and $\rho$ represents the density. The sum of each phase volume fraction is 1, which can be expressed as:

$$\sum_{q=1}^{n} \partial_q = 1 \tag{2}$$

The momentum equation is expressed as [32]:

$$\rho \frac{\partial v}{\partial t} + \rho v \cdot \nabla v = -\nabla p + \left[ \mu \left( \nabla v + \nabla v^T \right) \right] + \rho g + F \tag{3}$$

In the formula, $F$ represents the external force applied.

### 2.2.2. DPM Model Theory

The volume fraction of sand particles in the water body is very small, so the collision between particles is not considered in the analysis of sediment erosion. In the fluid, the force of particles can be expressed as [33,34]:

$$\frac{du_p}{dt} = F_D(u - u_p) + \frac{g(\rho_p - \rho)}{\rho_p} + F_y \tag{4}$$

$$F_D = \frac{18\mu C_D \text{Re}_p}{24\rho_p d^2{}_p} \tag{5}$$

$$\text{Re}_p = \frac{\rho d_p |u - u_p|}{\mu} \tag{6}$$

$$C_D = \alpha_1 + \frac{\alpha_2}{\text{Re}_p} + \frac{\alpha_3}{\text{Re}_p} \tag{7}$$

In the formula, $u_p$ is the particle velocity, $\rho$ is the fluid density, $\rho_p$ is the particle density, and $F_y$ is the force in the Y direction. $C_D$ is the drag coefficient, and $\text{Re}_p$ is the relative Reynolds number. $d_p$ is particle diameter, $u$ is the fluid velocity and $\mu$ is the hydrodynamic viscosity; $\alpha_1$, $\alpha_2$ and $\alpha_3$ are constants. The Erosion model agreed to the Erosion/Accretion model.

The Erosion amount of this model is related to fluid velocity and particle parameters, and its Erosion model equation is [35]:

$$\text{Re} = \sum_{p=1}^{N_p} \frac{m_p C(d_p) f(\alpha) u_p^{b(v)}}{A_{face}} \tag{8}$$

In the formula, $Re$ is the wall wear, $N_p$ is the number of particles, $m_p$ is the mass of particles, $C(d_p)$ is the particle diameter function, $F(\alpha)$ is the invasion angle function, $u_p$ is the velocity of particles relative to the wall, and $b(v)$ is the relative velocity function. In general, the empirical constant is 2.41, and $A_{face}$ is the wall area of the calculation region.

### 2.3. Mesh Partitioning and Boundary Conditions

Modeling and meshing were carried out for the spillway fluid area, and the mesh-independent verification showed that when the number of meshing approached 10,000, the obtained erosion value tended to be stable. Considering the accuracy and efficiency of the simulation, the obtained finite element model of the fluid region is shown in Figure 2. The finite element model consists of 106,081 elements and 112,278 nodes.

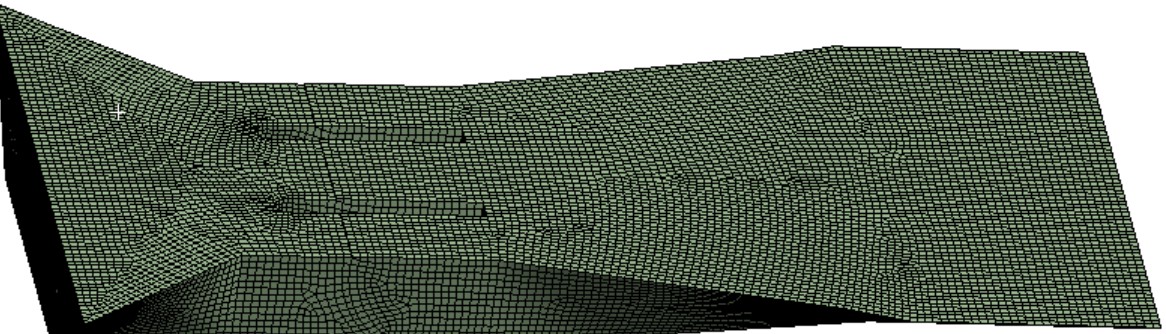

**Figure 2.** Finite element model of fluid region.

The medium inside the fluid area is water and air; the turbulence model is the standard $k\text{-}\omega$ model, the inlet is set as the speed inlet, the outlet is set as the pressure outlet, the outlet pressure is set as the standard atmospheric pressure, the surface of the gate pier and the spillway wall is set as the standard wall. The particles were set as inert particles in the DPM model, and the particle flow was injected from the inlet of the fluid domain along the normal surface direction, the boundary set as reflect. The Erosion/Accretion model was opened in the DPM physical model.

## 3. Result and Analysis

### 3.1. Spillway Erosion Analysis

The fluid area of the spillway was simulated and analyzed. The velocity of the fluid and discrete phase particle flow was set as 5 m/s, particle diameter was 0.025 mm, particle concentration was 3.2 kg/m$^3$, opening degree of the three radial leaves of the spillway were 2 m, and the water level before the radial leaf was 6.5 m. In the simulation process, the discrete phase is opened after the flow is stable, and the distribution of spillway erosion is observed after 20 s of particle flow erosion. The erosion amount of each part of the spillway obtained from simulation analysis is shown in Figure 3. As can be seen from Figure 3a, serious erosion at the bottom of the spillway is mainly distributed gate chamber section and the front part of the middle section of the spillway, while the erosion at the front section and the rear section of the spillway is very slight. Figure 3b shows the erosion situation of the spillway wall, and there is no obvious large-scale erosion phenomenon on the spillway wall. Figure 3c shows the erosion of the gate pier in the spillway, and there is no obvious erosion of the gate pier. Figure 3d shows the erosion of the gate leaf in the spillway, and the erosion is obvious and evenly distributed on the curved gate leaf.

Figure 4a shows that there is significant erosion wear on the gate chamber section and the front part of the middle section of the spillway. Figure 4b,c shows that there are no obvious erosion on the spillway wall and the spillway pier. Due to the lack of detailed information on the erosion of spillway by sediment during the actual operation of the gate, this paper only makes a quantitative analysis when comparing the simulation results with the actual situation. Because the erosion wear parts of the simulation results are the same as the actual situation, it can be considered that the simulation results are consistent with the erosion wear situation in the actual hydraulic construction operation process. Considering the actual erosion situation and the important role of the gate in the spillway for water storage and flood discharge, the key parts to be considered in the study of the spillway erosion are the gate leaf, the gate chamber section and the front part of the middle section of the spillway.

(**a**) Erosion cloud diagram of spillway bottom.

(**b**) Erosion cloud diagram of spillway wall.

(**c**) Erosion cloud diagram of spillway pier.　　　(**d**) Erosion cloud diagram of gate leaf.

**Figure 3.** Erosion cloud diagram of various parts of spillway.

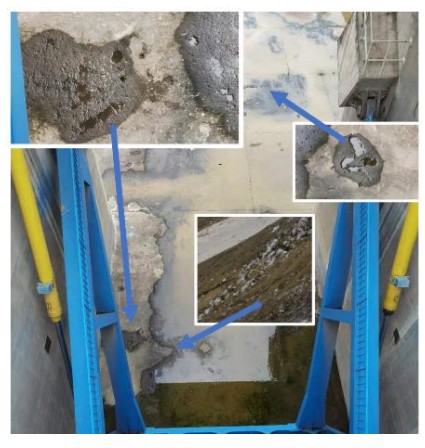

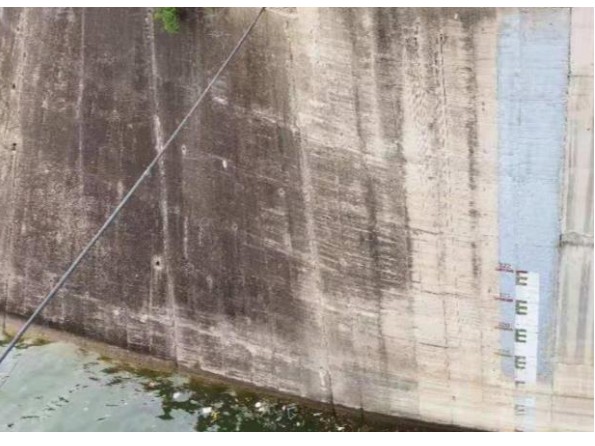

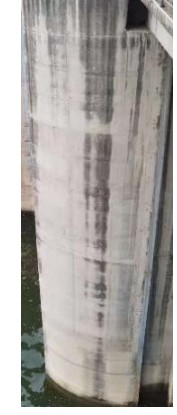

(**a**) the bottom of the rear　　　　(**b**) the sides of the spillway walls　　　(**c**) the gate pier portion of the spillway chamber

**Figure 4.** Erosion wear of spillway.

### 3.2. Effect of Erosion under Different Conditions

The influence of different factors on the spillway was further considered. According to the analysis results in 2.1, the main erosion positions of the spillway occurred at the bottom of the gate chamber and the middle section of the spillway. This part suppressed part of the flow field in Figure 1, and only the erosion conditions of the middle gate chamber and the corresponding middle section of the spillway under different conditions were analyzed. The finite element model of fluid domain in middle gate chamber and corresponding middle gate section is shown in Figure 5.

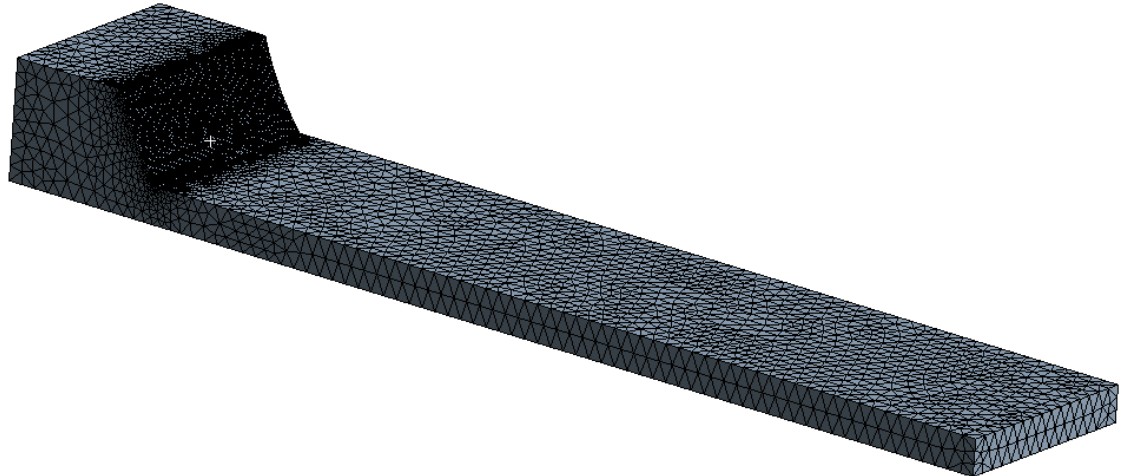

**Figure 5.** Finite element model of fluid domain in middle gate chamber and corresponding middle gate section.

In the process of spillway drainage, the erosion of the spillway by sediment is affected by many factors. Based on the analysis of the hydrological characteristics of the sediment flow and the working characteristics of the curved gate, the effects of sediment particle size, sediment concentration, flow velocity and gate opening degree on the spillway erosion are only considered in this paper.

#### 3.2.1. Effect of Particle Size on Corrosion

The sand particle diameter is an important factor affecting the erosion wear of the spillway caused by sand-laden water flow. In this paper, the effects of sediment with different particle sizes $d$ of 0.005 mm, 0.01 mm, 0.015 mm, 0.02 mm, 0.025 mm, 0.03 mm, 0.035 mm and 0.04 mm on the erosion amount of spillway are simulated. In consideration of the impact of sand particle size on spillway erosion, the sediment concentration in water is 3.2 kg/m$^3$, the water depth before setting the spillway is 6.5 m, the opening degree of the gate is 1 m, and the flow velocity is 5 m/s. The simulation process is that the multiphase flow model is opened after the water flow is stable, and the erosion cloud images of the bottom of the spillway and the gate leaf five seconds later are shown in Figures 6 and 7. Obviously, when only sediment diameter changes, the position of maximum erosion at the bottom of the spillway remains unchanged, which is in front of the ground sill (the position of the ground sill is indicated in Figure 6), and the erosion position is mainly concentrated around the ground sill. With the increase in sediment diameter, the erosion of gate leaf tends to be evenly distributed.

The relationship between the maximum erosion amount and sediment diameter is shown in Figure 8. When the particle diameter varies from 0.005 mm to 0.04 mm, the relationship between sand particle diameter and erosion amount is not monotonic: The erosion amount increases with the increase in particle size when the sediment diameter is 0.005 mm~0.015 mm, and decreases with the decrease in particle size when the sediment diameter is 0.015 mm~0.04 mm.

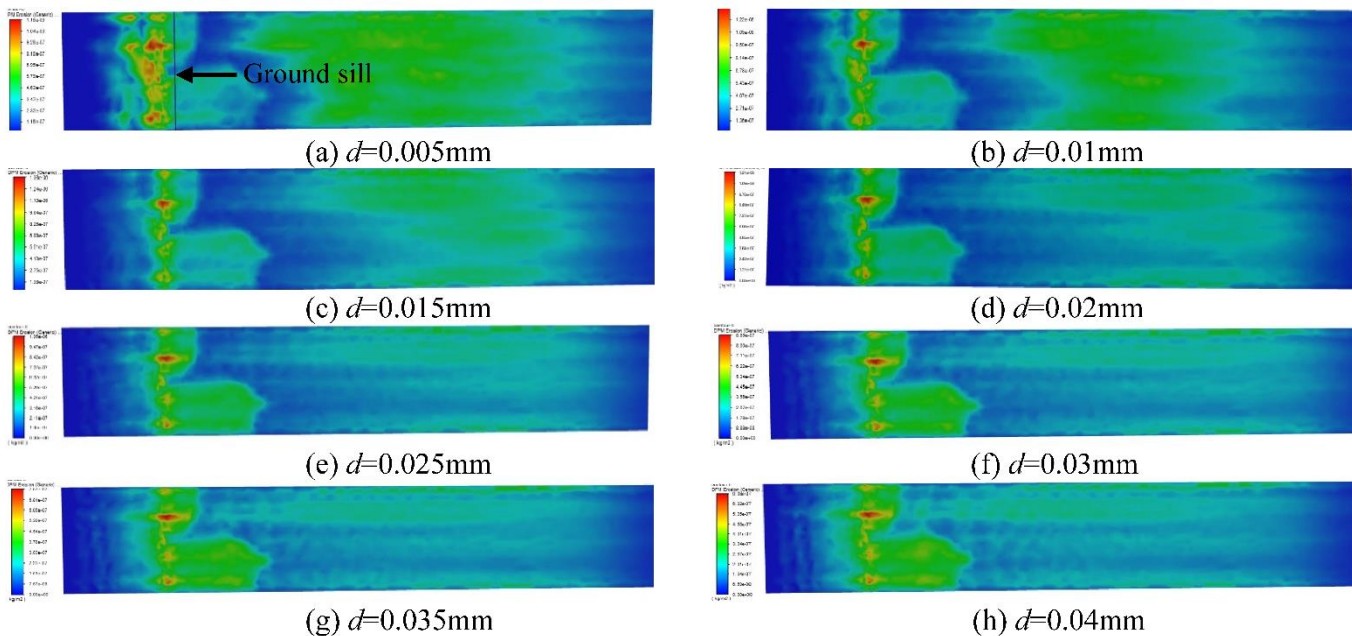

**Figure 6.** Erosion cloud diagram of spillway bottom under sediment with different diameters.

This result is inconsistent with the experimental results of Xiong [9], whose research results show that under the condition of consistent velocity, the particles with large particle sizes will produce greater erosion amounts on the erosion surface. However, when considering the effect of water flow, particle size will affect the carrying capacity of water flow on particles, so that sediment particles with different particle sizes can obtain different velocities under the condition of the same flow velocity. According to the velocity distribution of particle flow in Figure 9, when the particle diameter is 0.008 mm, the maximum velocity of particle flow is 15.1 m/s, while when the particle diameter is 0.04 mm, the maximum velocity of particle flow is only 10.8 m/s. When the sand particle diameter is larger, the mass of a single sand particle is also larger, and the carrying effect of water flow on sand particles is significantly reduced, leading to a lower velocity of particle flow. Therefore, when the sand particle size increases to a certain range, the erosion amount will decrease with the increase in the sand particle size.

### 3.2.2. Effect of Sediment Concentration on Erosion

The sediment concentration in the spillway reservoir area is greatly affected by the season, and the sediment concentration is also an important index affecting erosion. Combined with the actual situation, six different conditions of sediment concentration of 1 kg/m$^3$, 2 kg/m$^3$, 3 kg/m$^3$, 4 kg/m$^3$, 5 kg/m$^3$, 6 kg/m$^3$ were taken for simulation analysis. Sediment particle size is 0.025 mm, the water depth before the spillway is set at 6.5 m, the opening degree of the gate is 1 m, and the flow velocity is 5 m/s. The shapes of erosion cloud diagrams corresponding to different sediment concentrations are consistent, and the erosion conditions are basically the same as those in Figures 6e and 7e, except that the amount of erosion varies with different sediment concentrations. The maximum value of erosion amount corresponding to different sediment concentrations is shown in Figure 10. The value of erosion amount shows an obvious linear relationship with the increase in sediment concentration, and this distribution rule is also obviously consistent with the conclusion of Hu [6].

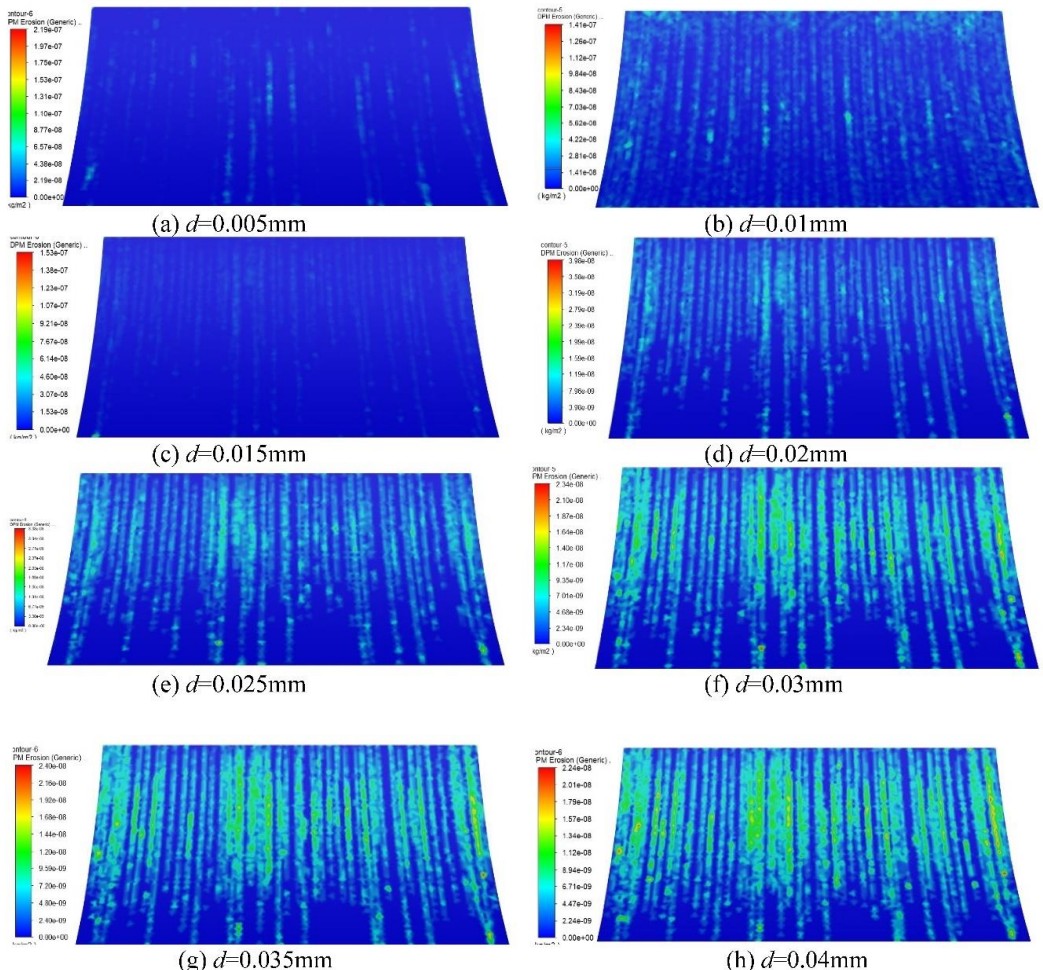

(a) *d*=0.005mm

(b) *d*=0.01mm

(c) *d*=0.015mm

(d) *d*=0.02mm

(e) *d*=0.025mm

(f) *d*=0.03mm

(g) *d*=0.035mm

(h) *d*=0.04mm

**Figure 7.** Erosion cloud diagram of gate leaf under sediment of different diameters.

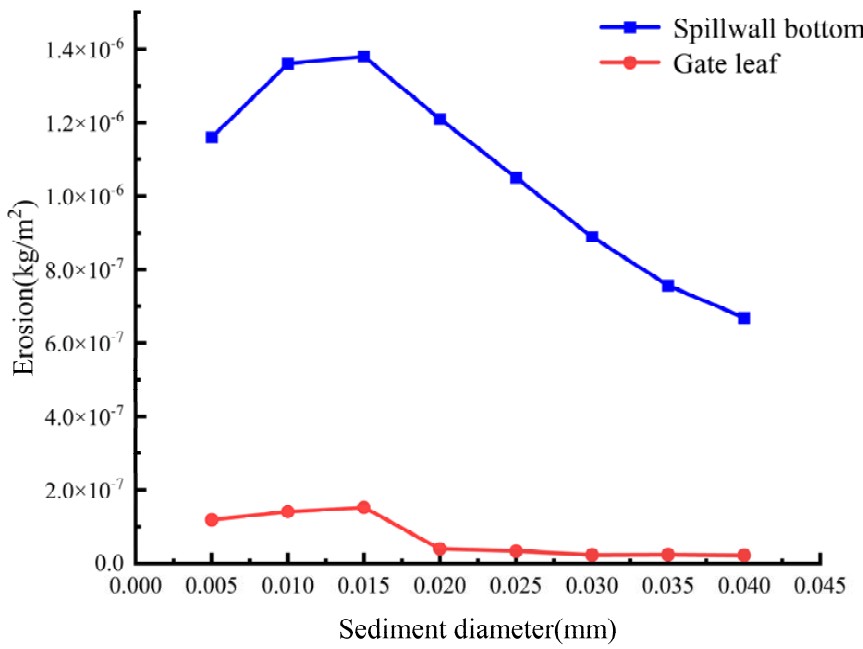

**Figure 8.** Relationship between erosion and sediment diameter.

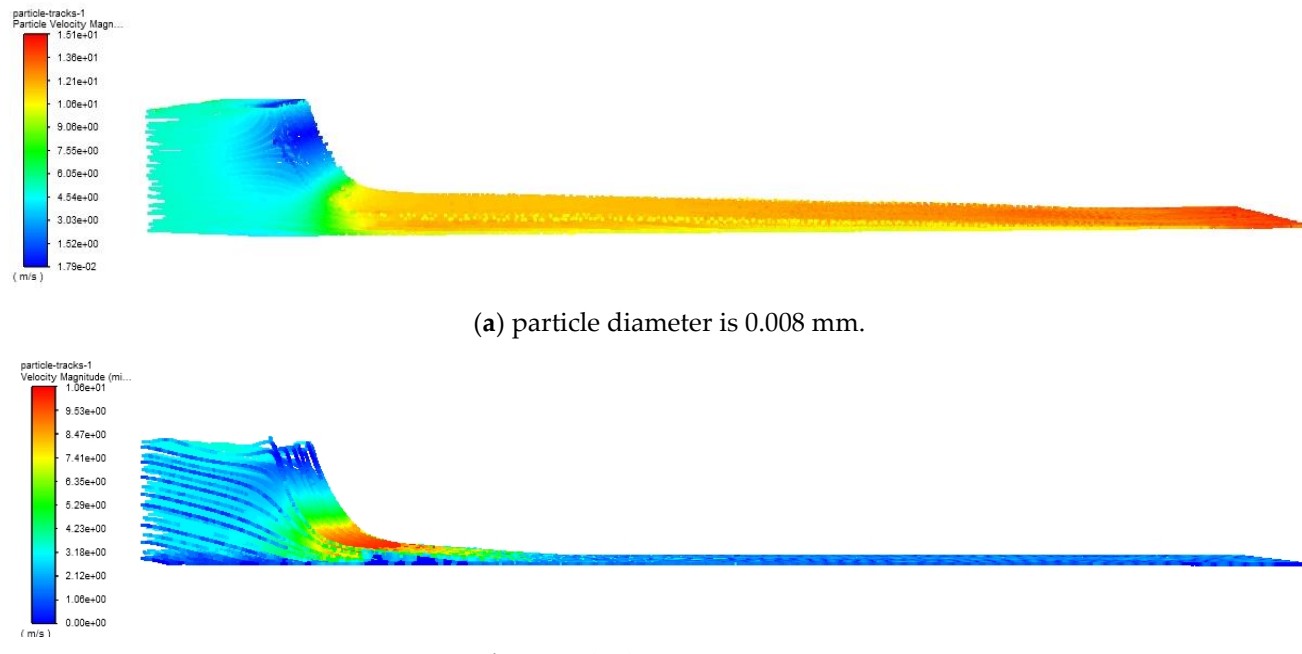

(**a**) particle diameter is 0.008 mm.

(**b**) particle diameter is 0.04 mm.

**Figure 9.** Cloud diagram of particle velocity distribution under different particle sizes.

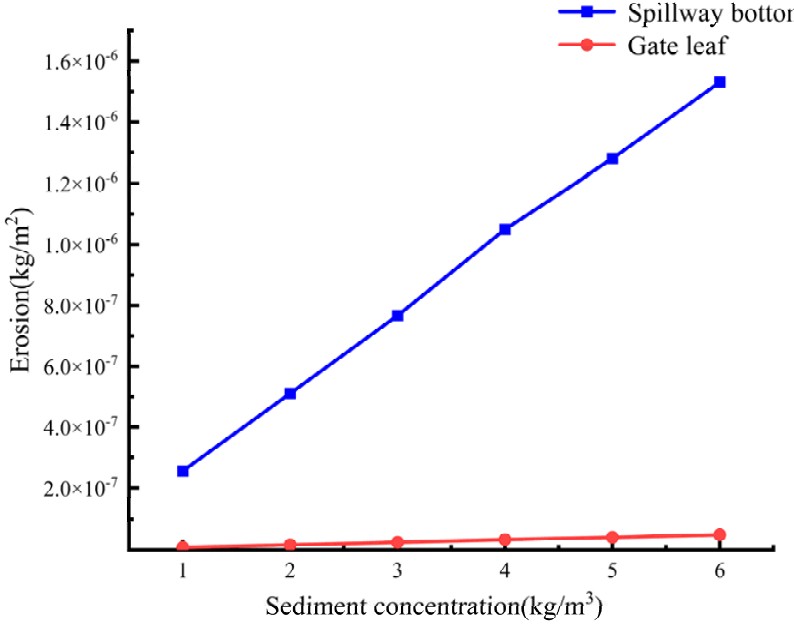

**Figure 10.** Relationship between erosion and sediment concentration.

### 3.2.3. Effect of Flow Velocity on Erosion

Under the condition of other factors being the same, the influence of the inlet velocity of the spillway on erosion was analyzed. The inlet velocity of the fluid domain is 2 m/s, 3 m/s, 4 m/s, 5 m/s, 6 m/s, 7 m/s and 8 m/s, 9 m/s. The sediment concentration is 3.2 kg/m$^3$, the sediment particle size is 0.025 mm, the water depth before setting the spillway is 6.5 m, and the opening degree of the gate is 1 m. The cloud diagram of the bottom erosion of spillway corresponding to different flow velocities is shown in Figure 11. Under different inlet flow velocities, the maximum erosion area of the spillway remains unchanged and is always in front of the ground sill, while the most obvious erosion area is also distributed around the ground sill. As the inlet flow velocity increases, the bottom

erosion area of the spillway becomes larger. When the speed is greater than 7 m/s, there are also noticeable erosion areas in the rear part of the middle section. The erosion amount distribution of different flow rates corresponding to the gate leaf part is basically the same, and the erosion amount cloud diagram is basically the same as that of Figure 7e.

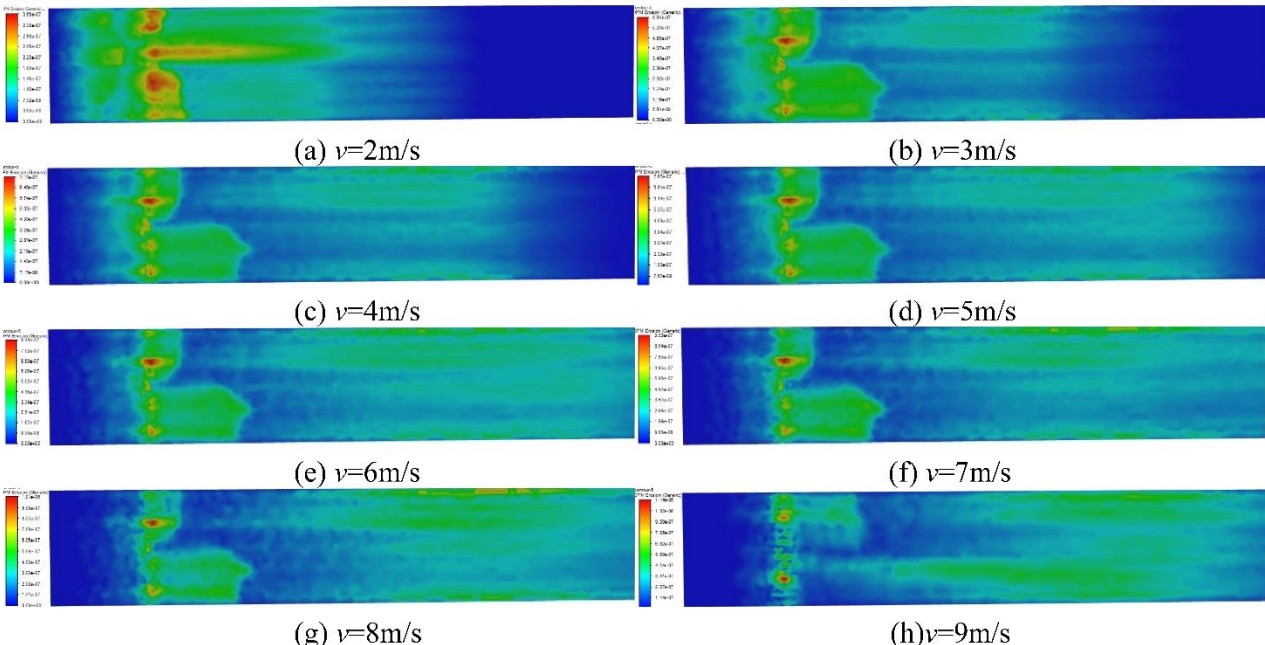

(a) *v*=2m/s (b) *v*=3m/s

(c) *v*=4m/s (d) *v*=5m/s

(e) *v*=6m/s (f) *v*=7m/s

(g) *v*=8m/s (h)*v*=9m/s

**Figure 11.** Erosion cloud diagram of spillway bottom at different flow rates.

The maximum erosion amount corresponding to different inlet flow velocities is shown in Figure 12. In general, the erosion amount increases with the increase in flow velocity, but the growth trend is more complex. For the bottom of the spillway, the erosion increment decreases with the increase in the flow velocity. When the velocity is greater than 7 m/s, the erosion increment does not increase much and tends to be stable. For the gate, the erosion increases exponentially with the increase in flow velocity.

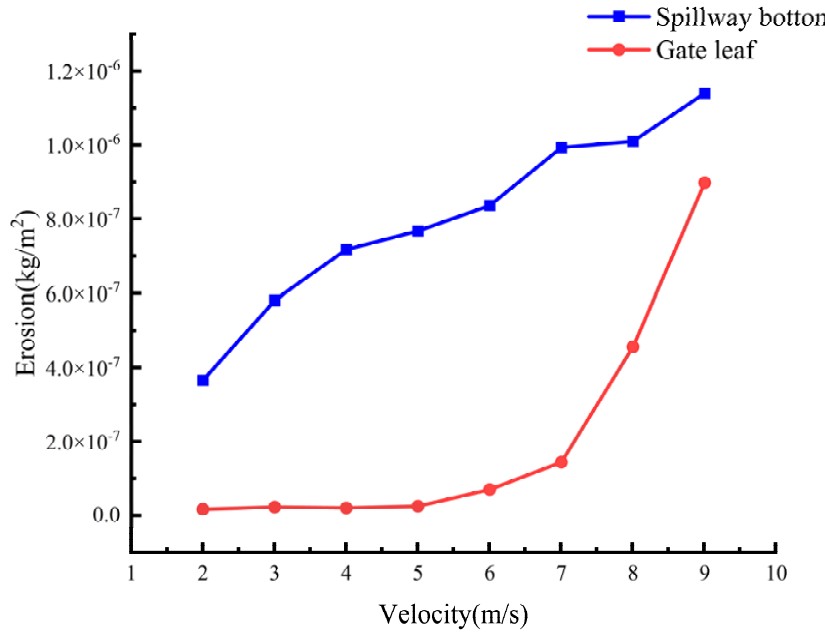

**Figure 12.** Relationship between erosion amount and inlet velocity of flow field.

The relationship between erosion and flow velocity at the bottom of the spillway and gate leaf is complicated. The relationship between erosion velocity and flow velocity is analyzed from particle energy and particle velocity vector. For particles, the kinetic energy equation is:

$$e_k = \frac{1}{2} m_p v^2 \tag{9}$$

where, $e_k$ is the kinetic energy of the particle, $m_p$ is the mass of the particle, and $v$ is the velocity value of the particle. When particles impact, their own kinetic energy will affect the erosion amount. However, the relationship between kinetic energy and velocity is a quadratic function, so the change rule of erosion amount and velocity cannot be explained from the perspective of kinetic energy alone. From the perspective of particle movement, the impact of sediment particles on the face plate and the bottom of the spillway is shown in Figure 13. For the impact angle of gate leaf $\alpha_1$, the gate leaf radian is ignored, and the impact angle is as follows:

$$\alpha_1 = \arctan \frac{v_y}{v_z} = \frac{v_0 + v'}{gt} \tag{10}$$

The impact angle $\alpha_2$ at the bottom of the spillway is expressed as:

$$\alpha_2 = \arctan \frac{v_z}{v_y} = \arctan \frac{gt}{v_0 + v'} \tag{11}$$

where, $v_0$ represents the initial velocity of the sand inlet, $v_y$ and $v_z$ represent the velocity components of the sand in horizontal and vertical directions, respectively, represent the velocity increment caused by water carrying action, $g$ is the acceleration of gravity, and T is the time value of the particle moving from the flow field inlet to the current position. As $v_0$ increases, $\alpha_1$ tends to be 90°, and $\alpha_2$ tends to be 0°. The study of Xiong et al. [9] shows that the erosion amount will change with the change in erosion angle. Therefore, the complex relationship between the maximum erosion amount of the gate and the inlet velocity of the flow field is caused by the change in particle erosion velocity and impact angle.

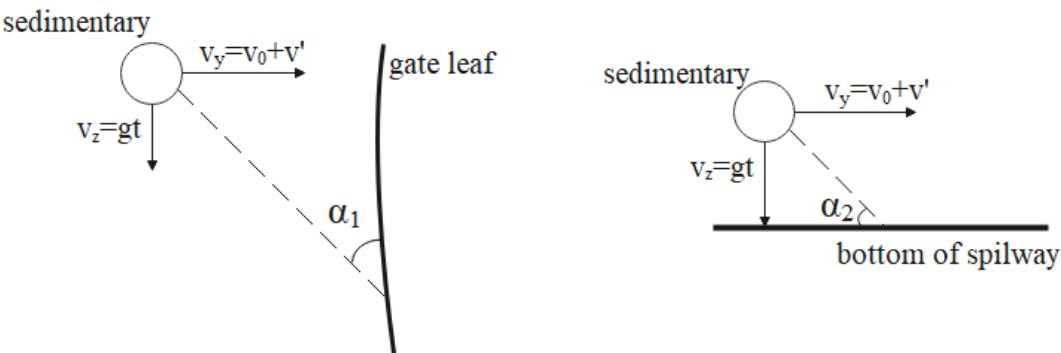

(**a**) Particle impact angle at gate leaf.　　　　(**b**) Particle impact angle at bottom of spillway.

**Figure 13.** Velocity component of particle impact.

Therefore, the flow velocity increases the velocity of sediment, impacting the spillway and also changes the incidence angle of sediment erosion. The conclusion of Xiong et al. [9] shows that both flow velocity and incident angle have an effect on erosion. When sand-bearing flow erodes the spillway, the inlet flow velocity simultaneously changes the erosion velocity and incident angle of sediment particles. It is the joint action of this factor that makes the relationship between erosion quantity and inlet flow velocity more complex.

### 3.2.4. Effect of Gate Opening on Erosion

Gate opening is one of the main parameters of gate operation. Gate opening will affect the characteristics of water flow in the spillway, so it will also affect the erosion wear.

Under the condition of water depth of 6.5 m, the erosion wear of spillway with the opening of 1 m, 2 m, 3 m, 4 m, 5 m and 6.5 m is simulated. The sediment concentration is 3.2 kg/m$^3$, the particle size of sediment is 0.025 mm, and the inlet velocity of the fluid domain is 5 m/s. The cloud diagram of spillway erosion results is shown in Figure 14. As the opening degree of the gate increases, the location where the maximum erosion occurs at the bottom of the spillway gradually moves backward, and the overall distribution of the erosion area has no obvious rule. The change in gate erosion with the opening is mainly reflected in that of the increase in the gate opening, the contact area between gate and water becomes smaller, and the corresponding erosion area becomes smaller.

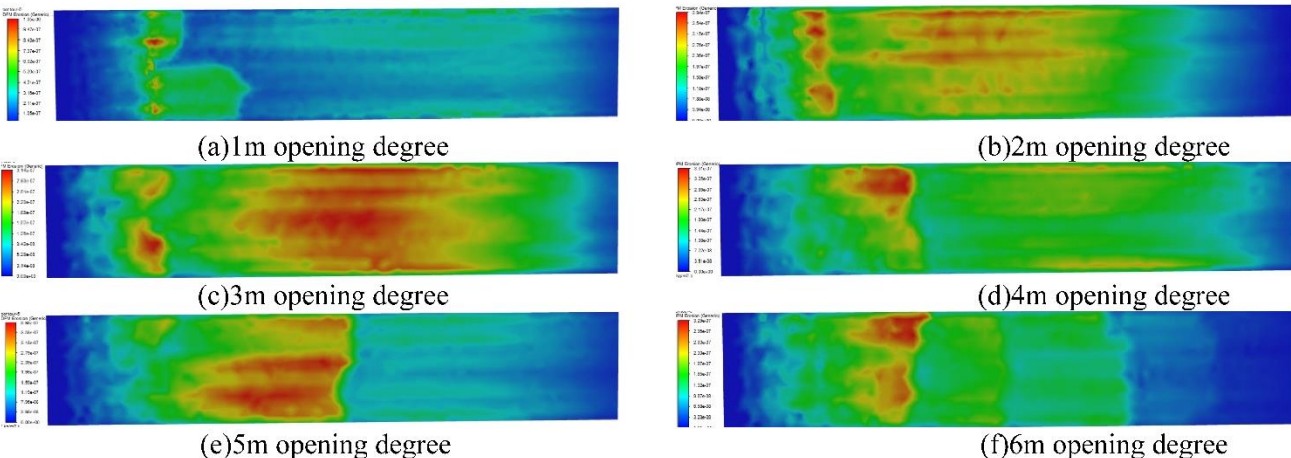

(a)1m opening degree　　　　　　　　　　(b)2m opening degree

(c)3m opening degree　　　　　　　　　　(d)4m opening degree

(e)5m opening degree　　　　　　　　　　(f)6m opening degree

**Figure 14.** Erosion cloud diagram of spillway with different gate openings.

The relationship between the maximum erosion amount and the opening of the gate is shown in Figure 15. The erosion amount at the bottom of the spillway has a negative linear correlation with the opening of the gate when the opening of the gate is between 1 m and 3 m and tends to be stable when the opening of the gate surface reaches 3 m.

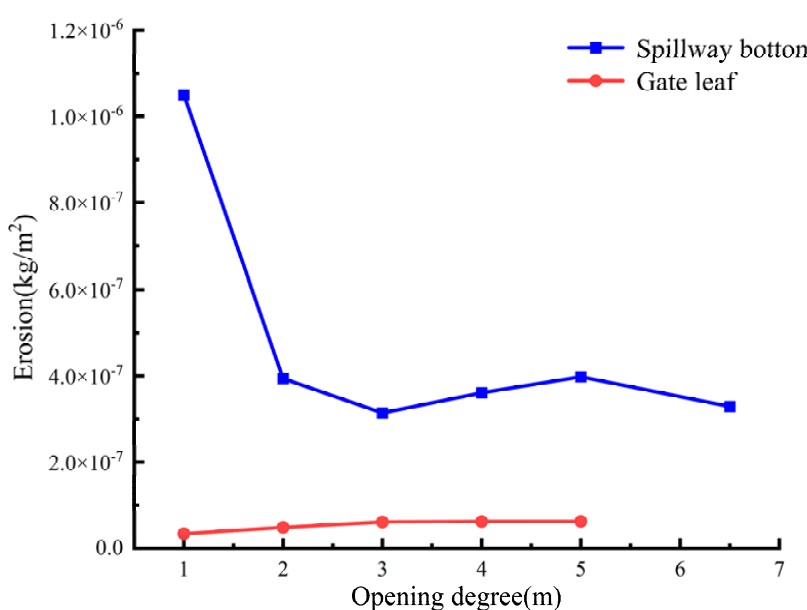

**Figure 15.** Relationship between erosion and gate opening.

The research objects of the other literature are all related to the influence of gate opening to offset erosion. According to the research results of Chen [36], gate opening will change the flow field velocity. As this paper concludes in Section 3.2.3, flow velocity

also affects erosion. As shown in Figure 16, the change in gate opening causes significant changes in the flow velocity around the gate. When the gate opening is 1 m, 3 m and 6.5 m, the corresponding flow velocity behind the spillway gate is 13.7 m/s, 11.1 m/s and 9.52 m/s, respectively. In essence, the change in gate opening causes the change of flow velocity behind the gate, which leads to the change in erosion amount.

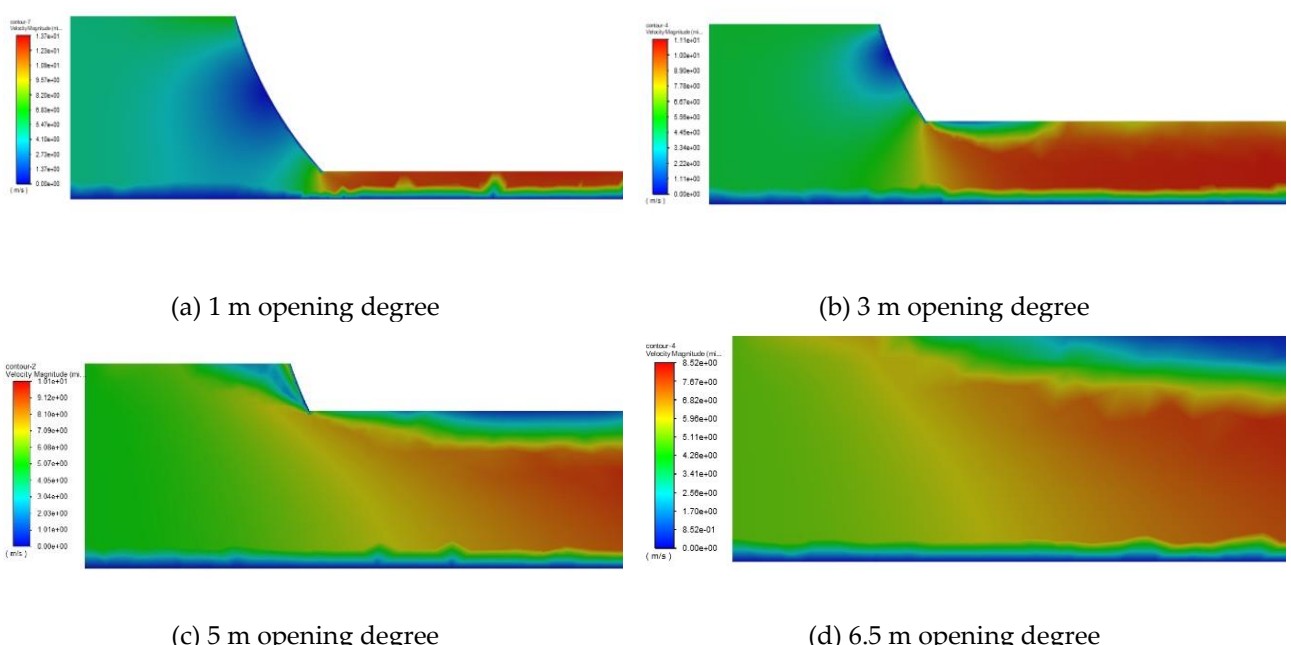

(a) 1 m opening degree

(b) 3 m opening degree

(c) 5 m opening degree

(d) 6.5 m opening degree

**Figure 16.** Flow field velocity near the gate with different opening degrees.

## 4. Conclusions

(1) The spillway erosion mainly occurs at the bottom of the spillway chamber and the middle section, and the most serious erosion wear occurs near the ground sill. Compared with the bottom of the spillway, the gate leaf wear is relatively slight, but due to the importance of the radial leaf in hydraulic construction, the erosion wear of the gate leaf should not be underestimated;

(2) Under the same other conditions, the erosion amount increases first and then declines with the increase in sediment particle size, and the maximum erosion amount increases linearly with the increase in sediment concentration;

(3) The maximum erosion of the spillway is positively correlated with the flow velocity. At the bottom of the spillway, the erosion increment decreases with the increase in the flow velocity. At the gate, the erosion increases exponentially with the increase in the flow velocity;

(4) The opening of the radial leaf will affect the amount and location of sediment erosion on the spillway. When the gate opening degree is small (0~3 m when the water level is 6.5 m), the erosion at the bottom of the spillway reduces with the increase in the gate opening. When the opening reaches a certain range, the maximum erosion at the bottom of the spillway is usually stable. At the same time, with the increase in the opening, the highest erosion area at the bottom of the spillway changes gradually from the bottom sill of the gate to the rear section of the gate;

(5) In this paper, a mechanism model of spillway erosion is built through numerical simulation, which can predict the erosion of the gate to a certain extent. Further research in this paper can be carried out by collecting information such as sediment diameter, sediment concentration, flow velocity and gate opening degree of the spillway at each working time. Then the erosion simulation model is used to simulate

the erosion of the spillway so as to realize more economical and effective operation and maintenance of the spillway.

**Author Contributions:** Formal analysis, Y.Z.; Investigation, C.Z.; Resources, T.W.; Software, M.W.; Supervision, Y.Z.; Writing—Original draft, C.Z.; Writing—Review & editing, C.Z. and H.Z. All authors have read and agreed to the published version of the manuscript.

**Funding:** This research was funded by Project of Integrated Standardization and New Mode Application of Intelligent Manufacturing of Ministry of Industry and Information Technology of China, grant number 2018037. This research was also funded by Water Conservancy Science and Technology Project of Henan Provincial Water Resources Department, grant number GG202068No.2018037.

**Institutional Review Board Statement:** Not applicable.

**Informed Consent Statement:** Not applicable.

**Data Availability Statement:** Not applicable.

**Conflicts of Interest:** The authors declare no conflict of interest.

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
