# Peer review of "Numerical Investigation of Erosion Wear Characteristics of Hydraulic Spillway"

_applsci, doi:10.3390/app11178118_

Round 1

Reviewer 1 Report

Dear Editor,

The manuscript "Numerical Investigation of Erosion Wear Characteristics of Hy-2 Hydraulic Spilway" was submitted to the applied sciences journal. It's nice and terrific research with the collaboration of Chinese researchers about assessment of erosion characteristics of Spillway. The topic is so valuable, but there are some significant concerns in this version that I can't accept. Finally, I present some suggestions for improving the quality of this MS as following:

  • Spillway or Spillway, which one? Please check more carefully.
  • It is recommended to re-write the abstract and highlight the quantitative results. The research findings are precious, it is recommended to mention a little more quantitative results in the abstract.
  • There is no conclusion in the abstract. This research should have an exciting conclusion to attract many readers.
  • It is recommended that there is no similarity between the title and the keywords, so more people will access your article in their searches through scientific databases.
  • In the introduction, it is recommended to use more references. In this section, 16 references have been used and should be increased to 25-26 items. Please, use references that have been very impressive in recent years.
  • Use more researches on the literature review of VOF and DPM models.
  • The results section is well written, congratulations.
  • I can't find and discussion section!!!! Please be careful. I'm such concern about the lack of this section. It's essential.
  • One of the significant concerns is that the authors should carefully develop a discussion section to talk about the significance, shortages or advantages of the methods you proposed, the reliability and meaning of your results (compared to other related studies) etc.
  • It is recommended to increase the number of references.
  • Please be sure that all the references cited in the manuscript are also included in the reference list and vice versa with matching spellings and dates.
  • I checked plagiarism detection of this research and the similarity is 13% and there is not any problem, please check the attached file.
  • Finally, It is needed that it be read by a native author or an English institute, it is a constructive suggestion.

Best Regards

Author Response

Authors would like to thank reviewers for their valuable comments and suggestions, which have helped us to improve the manuscript. The authors have studied all comments carefully and made all possible corrections accordingly. Please find the answers to the comments given below;

  • Spillway or Spillway, which one? Please check more carefully.

Spillway. I apologize for my spelling mistakes. I have checked and corrected in the manuscript. 

  • It is recommended to re-write the abstract and highlight the quantitative results. The research findings are precious, it is recommended to mention a little more quantitative results in the abstract.

Thank you for your good comments. Based on the comments of the reviewers, we have reworked the abstract.

  • There is no conclusion in the abstract. This research should have an exciting conclusion to attract many readers.

The author has recognized this problem and added a conclusion section at the end of the abstract.

  • It is recommended that there is no similarity between the title and the keywords, so more people will access your article in their searches through scientific databases.

Introduction has been revised. Hydraulic structures was added as a new keyword.

  • In the introduction, it is recommended to use more references. In this section, 16 references have been used and should be increased to 25-26 items. Please, use references that have been very impressive in recent years.

Introduction has been revised. References have been increased to 26 items in the introduction.

  • Use more researches on the literature review of VOF and DPM models.

      Introduction has been revised.There are more researches on the literature review of VOF and DPM models are used.

  • The results section is well written, congratulations.
  • I can't find and discussion section!!!! Please be careful. I'm such concern about the lack of this section. It's essential.
  • One of the significant concerns is that the authors should carefully develop a discussion section to talk about the significance, shortages or advantages of the methods you proposed, the reliability and meaning of your results (compared to other related studies) etc.

 It has been revised in the paper according to the reviewer’s suggestion.The discussion was added to the manuscript,and at the end of the article the author expressed his views on the significance of the article.

  • It is recommended to increase the number of references.

The authors have added 14 references, bringing the total to 36.

  • Please be sure that all the references cited in the manuscript are also included in the reference list and vice versa with matching spellings and dates.

The author cross-checks and ensures that all references are in the literature list, and vice versa with matching spellings and dates

  • I checked plagiarism detection of this research and the similarity is 13% and there is not any problem, please check the attached file.
  • Finally, It is needed that it be read by a native author or an English institute, it is a constructive suggestion.

According to the reviewer's good instruction, we have revised the whole manuscript carefully and tried to avoid any grammar or syntax error.

The revised manuscript is attached.

Reviewer 2 Report

The manuscript cannot be accepted for publication in its current format.

The English of the manuscript is poor and must be improved before resubmission. 

Author Response

Authors would like to thank reviewers for their valuable comments and suggestions, which have helped us to improve the manuscript. The authors have studied all comments carefully and made all possible corrections accordingly. Please find the answers to the comments given below;

Reviewer #2

The manuscript cannot be accepted for publication in its current format.The English of the manuscript is poor and must be improved before resubmission.

The author has carefully read and revised the literature, and asked professionals to proofread it. As the author's first English paper, the author has tried his best to revise it.  If there are still any deficiencies in the manuscript,please the reviewer to give criticism and correction.  Thank you again for your valuable advice.

The revised manuscript is attached.

Reviewer 3 Report

This article presents an interesting argument: the effects caused by the diversion of sand-laden water on the wear of hydraulic structures, also considering a possible link with flood events and the expected increase of these events in the coming decades.

A more detailed description of the study area is necessary for a better understanding of the work: a specific section could be added. Of course, increase the number of references can strengthen the analysis carried out.

Specific comments are reported in the attached file.

Round 2

Reviewer 1 Report

Dear editor

I have reviewed the article and the authors have made great efforts to address my concerns, so in my opinion, it can be published in the Applied Sciences journal.

Reviewer 2 Report

The English of the manuscript is still weak and needs significant improvement.

Other than that, the manuscript is scientifically sound and robust, and can be published after minor revisions.